# Molecular mass growth through ring expansion in polycyclic aromatic hydrocarbons via radical–radical reactions

Long Zhao [1], Ralf.I. Kaiser [1], Wenchao Lu [2], Bo Xu [2], Musahid Ahmed [2], Alexander N. Morozov[3], Alexander M. Mebel [3], A. Hasan Howlader [3] & Stanislaw F. Wnuk [3]

Polycyclic aromatic hydrocarbons (PAHs) represent key molecular building blocks leading to carbonaceous nanoparticles identified in combustion systems and extraterrestrial environments. However, the understanding of their formation and growth in these high temperature environments has remained elusive. We present a mechanism through laboratory experiments and computations revealing how the prototype PAH—naphthalene—can be efficiently formed via a rapid 1-indenyl radical—methyl radical reaction. This versatile route converts five- to six-membered rings and provides a detailed view of high temperature mass growth processes that can eventually lead to graphene-type PAHs and two-dimensional nanostructures providing a radical new view about the transformations of carbon in our universe.

[1] Department of Chemistry, University of Hawaii at Manoa, Honolulu, HI 96822, USA. [2] Chemical Sciences Division, Lawrence Berkeley National Laboratory, Berkeley, CA 94720, USA. [3] Department of Chemistry and Biochemistry, Florida International University, Miami, FL 33199, USA. Correspondence and requests for materials should be addressed to R.I.K. (email: ralfk@hawaii.edu) or to M.A. (email: mahmed@lbl.gov) or to A.M.M. (email: mebela@fiu.edu)

Since the discovery of carbon-based molecules and dust in the interstellar medium, humans have been fascinated by what processes lead to their formation. However, the elucidation of the fundamental reaction pathways to polycyclic aromatic hydrocarbons (PAHs)—organic molecules composed of fused benzene rings with naphthalene ($C_{10}H_8$) denoting the simplest representative—in combustion environments[1] and in the interstellar medium (ISM)[2] has posed a long-standing challenge. Sophisticated combustion[3] and astrochemical reaction networks[4] along with flame sampling studies[5] and astronomical surveys[2] reveal that PAHs represent the critical link between resonantly stabilized free radicals (RSFRs) like propargyl ($C_3H_3^\bullet$), allyl ($C_3H_5^\bullet$), and cyclopentadienyl ($C_5H_5^\bullet$)[6,7] and carbonaceous nanoparticles universally referred to as soot and interstellar grains on Earth and in extraterrestrial environments, respectively[2,8,9]. Whereas on Earth, PAHs represent undesirable toxic, often carcinogenic byproducts released in incomplete combustion[10], in the interstellar medium, PAHs embody up to 20% of the galactic carbon budget[11] and are—along with carbonaceous grains—suggested to play a central role in the formation of vital precursors to molecular building blocks of life such as amino acids[8]. The firm identification of PAHs in carbonaceous chondrites such as Allende along with carbon isotopic analyses manifests an origin of PAHs in circumstellar envelopes of carbon stars involving extensive molecular mass growth processes[2,12,13]. The underlying astrochemical models rely predominantly on the Hydrogen-Abstraction/aCetylene-Addition (HACA) mechanism[14,15] suggesting molecular mass growth processes via stepwise ring addition to PAHs.

However, recent combustion models invoking the HACA mechanism reveal that successive additions of acetylene to PAHs as complex as coronene ($C_{24}H_{12}$) are inefficient and too slow to reproduce the quantified mass fractions and formation speeds of complex PAHs in combustion flames[16]. Moreover, HACA occurring at zigzag edges of PAH molecules tend to form only an extra five-membered ring instead of a six-membered one[17]. Most critically, in deep space, PAHs are rapidly destroyed by photolysis, galactic cosmic rays, and shock waves resulting in lifetimes of only a few $10^8$ years[18]. These time scales are significantly shorter than those for injection of PAHs into the interstellar medium by carbon rich Asymptotic Giant Branch (AGB) stars of $2 \times 10^9$ years[18]. Hence, PAH(-like) species should not exist in the interstellar medium, however the presence of such species indicates a central, hitherto elusive high-temperature route to rapid growth of PAHs in circumstellar envelopes of carbon rich stars.

Here, we report on a combined experimental and computational study on the reaction of the aliphatic methyl radical ($CH_3^\bullet$) with the aromatic 1-indenyl radical ($C_9H_7^\bullet$) leading eventually to the formation of naphthalene ($C_{10}H_8$)—the prototype PAH carrying two fused benzene rings. The reaction is enabled by the methyl radical—formed via pyrolysis of acetone ($(CH_3)_2CO$)—with the 1-indenyl radical—generated via pyrolysis of 1-bromoindene ($C_9H_7Br$)—in a chemical micro reactor[19] to form naphthalene ($C_{10}H_8$) together with two hydrogen atoms (H) via methylindene ($C_9H_7CH_3$; $C_{10}H_{10}$) intermediates (reaction (1a)). Electronic structure calculations support the experimental results to reveal that D3-methyl ($CD_3^\bullet$) (reaction (1b)), naphthalene formation is initiated by a barrierless radical–radical recombination leading to methylindene ($C_{10}H_{10}$) followed by atomic hydrogen loss, extensive isomerization of methylindenyl radicals ($C_{10}H_9^\bullet$) via hydrogen shifts and ring expansion, and termination through a second hydrogen loss forming naphthalene ($C_{10}H_8$) (Fig. 1). Although the formation of naphthalene via recombination of two cyclopentadienyl radicals ($C_5H_5^\bullet$)[20,21] and methylation of indenyl has been predicted theoretically[22,23], similar to the formation of benzene via methylation of the cyclopentadienyl

radical[24–26], the validity of elementary reactions between free hydrocarbon radicals leading to PAHs at high temperatures have not been realized experimentally. Indeed there is no experiment to date under high temperature conditions that has followed the kinetics and mechanism of a hydrocarbon free radical with a second hydrocarbon radical, and hence the reaction route through methylation of the indenyl radical provides a benchmark for the conversion of a five-membered ring to a six-membered ring in PAHs. Previous experimental attempts to probe elementary radical–radical reactions have been limited to low or room temperatures[27], in crossed beams environments[28] involving small species, such as the hydroxyl radical and atoms[29]. Not only is our experimental methodology crucial in modelling combustion and astrochemical processes, it also provides a strategy to study chemical reactions of radicals in general under high-temperature environments of relevance to synthesis and materials chemistry.

$$C_9H_7^\bullet + CH_3^\bullet \rightarrow C_9H_7CH_3 \rightarrow C_{10}H_9^\bullet + H^\bullet \rightarrow C_{10}H_8 + 2H^\bullet \quad (1a)$$

$$C_9H_7^\bullet + CD_3^\bullet \rightarrow C_9H_7CD_3 \rightarrow C_{10}H_6D_3^\bullet + H^\bullet \rightarrow C_{10}H_6D_2 + H^\bullet + D^\bullet \quad (1b)$$

## Results

**Mass spectra results**. Once formed in the chemical micro reactor, the products were probed isomer-specifically through fragment-free photoionization in a supersonic molecular beam coupled to synchrotron-based mass spectrometry (Method). Representative mass spectra collected at a photon energy of 9.50 eV for the reaction of the (D3)-methyl radical with the 1-indenyl radical (reaction (1)) are presented in Fig. 2; reference data were also collected by only seeding 1-bromoindene in helium carrier gas. In the methyl–indenyl system, the data provide compelling evidence on the formation of molecules connected to molecular ions with mass-to-charge ($m/z$) ratios of 131, 130, 129, and 128 (Fig. 2b), which can be associated with $^{13}CC_9H_{10}$, $C_{10}H_{10}$, $C_{10}H_9/^{13}CC_9H_8$, and $C_{10}H_8$, respectively. These ions are absent in the control experiments (Fig. 2a and Supplementary Figs. 4 and 5). In the 1-bromoindene—helium control experiment, only ion counts for 196 ($C_9H_7{}^{81}Br^+$, precursor), 194 ($C_9H_7{}^{79}Br^+$, precursor), 117 ($^{13}CC_8H_8^+$, $^{13}C$-indene), 116 ($^{13}CC_8H_7^+$, $^{13}C$-indenyl radical; $C_9H_8^+$, indene), and 115 ($C_9H_7^+$, indenyl radical) are detected with signal at $m/z = 117$, 116, and 115 also observed in the 1-indenyl-methyl system (Supplementary Fig. 4). For the (D6-) acetone—helium control experiments, no signal of more than $m/z = 58$ (Supplementary Fig. 5a) and 64 (Supplementary Fig. 5b) was observed, respectively. Upon replacing the methyl radical by the D3-methyl radical, these ion counts shift to $^{13}CC_9H_7D_3$ (134 amu), $C_{10}H_7D_3$ (133 amu), $C_{10}H_7D_2/^{13}CC_9H_6D_2$ (131 amu), and $C_{10}H_6D_2$ (130 amu) (Fig. 2c); for a full interpretation of all ion counts, please refer to the Supplementary Information (Supplementary Figs. 6–9). Considering the molecular weight of the reactants and the products, we conclude that the $C_{10}H_{10}$ molecule is formed in reaction (1a) via the radical–radical recombination of methyl with 1-indenyl followed by stabilization of the adduct; the $C_{10}H_8$ product is then formed via two successive atomic hydrogen losses as outlined schematically in reaction (1a). Under our high temperature experimental conditions, helium-seeded 1-bromoindene precursor does not lead to the formation of any naphthalene (Fig. 2b).

**Photoionization efficiency spectra results**. The isomers with molecular formulae $C_{10}H_{10}$ and $C_{10}H_8$ formed in the reaction of the 1-indenyl radical with the methyl radical along with their partially deuterated counterparts in the 1-indenyl—D3-methyl system are identified by examination of the corresponding photoionization

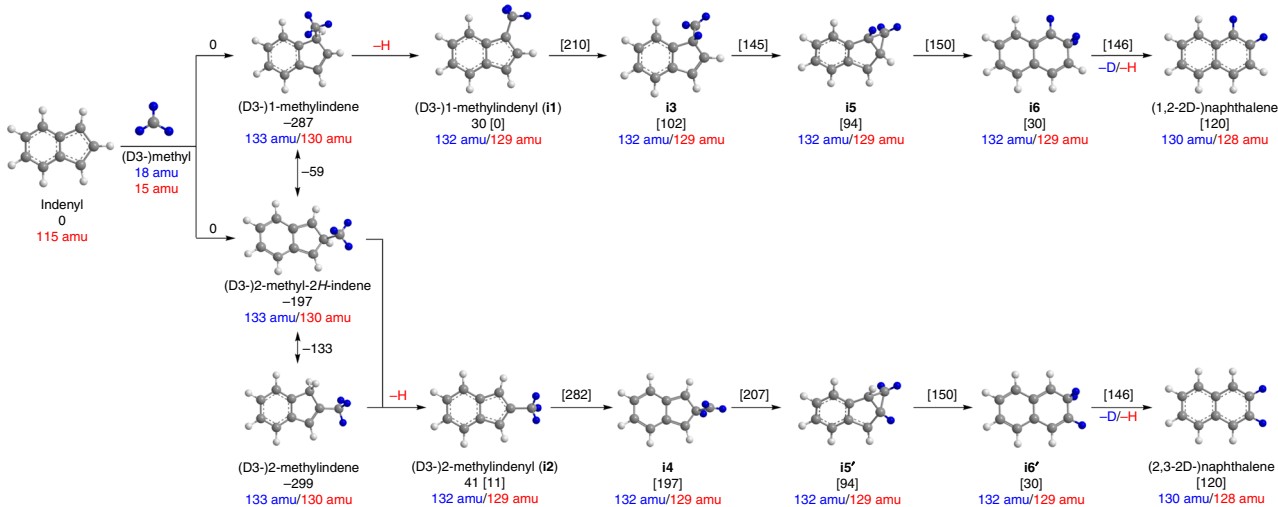

**Fig. 1** Reaction pathways for the 1-indenyl—(D3-)methyl systems leading to (D2-)naphthalene with atoms in blue tracing the deuterium atoms. The blue and red numbers indicate the mass-to-charge ratios of the reactants, intermediates, and products in the 1-indenyl/D3-methyl and 1-indenyl/methyl systems, respectively. The energies are defined with respect to the separated reactants; energies in square brackets are defined with respect to the 1-methylindenyl radical; all energies for the (partially) deuterated species are within 1-2 kJ mol$^{-1}$ of the non-deuterated counterparts. All energies are given in kJ mol$^{-1}$. The detailed potential energy surfaces and coordinates or reactants, intermediates, and products are compiled in the Supplementary Figs. 1 and 2, and Supplementary Note 2

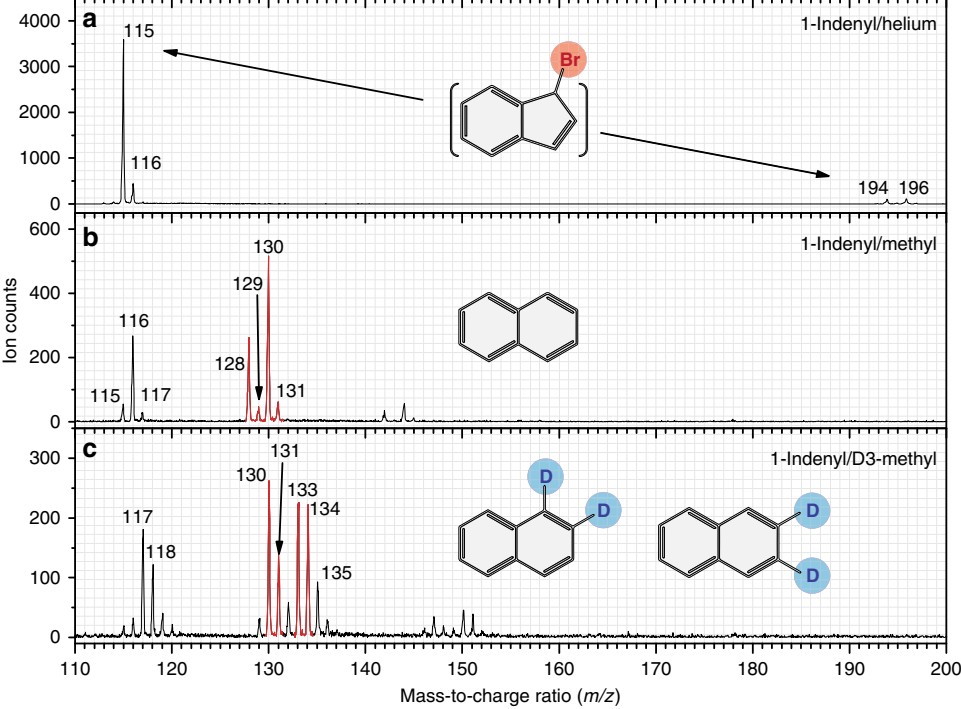

**Fig. 2** Mass spectra recorded at a photon energy of 9.50 eV and reactor temperature of 1425 ± 10 K. **a** 1-indenyl/helium, **b** 1-indenyl/ methyl, **c** 1-indenyl/ D3-methyl. The complete assignments of the ion peaks are presented in the Supplementary Information (Supplementary Figs. 6–9). The mass peaks identifying (partially deuterated) naphthalenes are marked in red. Source data are provided as a Source Data file

efficiency (PIE) curves, which display the intensity of the ions at a well-defined mass-to-charge ratio as a function of the photon energy from 7.30 to 10.00 eV (Fig. 3). These functions can be fit with established reference PIE curves for distinct $C_{10}H_{10}$ and $C_{10}H_8$ isomers. For the 1-indenyl-methyl system, the experimental PIE curve at $m/z = 128$ (Fig. 3a) can be reproduced with a reference PIE curve for naphthalene ($C_{10}H_8^+$). This PIE curve has an onset of the ion signal at 8.15 ± 0.05 eV, which correlates well with the adiabatic

ionization energy (AIE) of naphthalene of 8.12 ± 0.01 eV[30]. It is critical to stress that since the PIE curves of alternative $C_{10}H_8$ isomers like azulene, 3-buten-1-ynyl-benzene, and 1-buten-3-ynyl-benzene are fundamentally correlated to each isomer; the co-existence of alternative isomers in the molecular beam would alter the shape of the PIE considerably[31]. Therefore, we conclude that naphthalene is the sole contributor to signal at $m/z$ of 128 within our error limits. The PIE curves at $m/z = 128$ and 129 do not

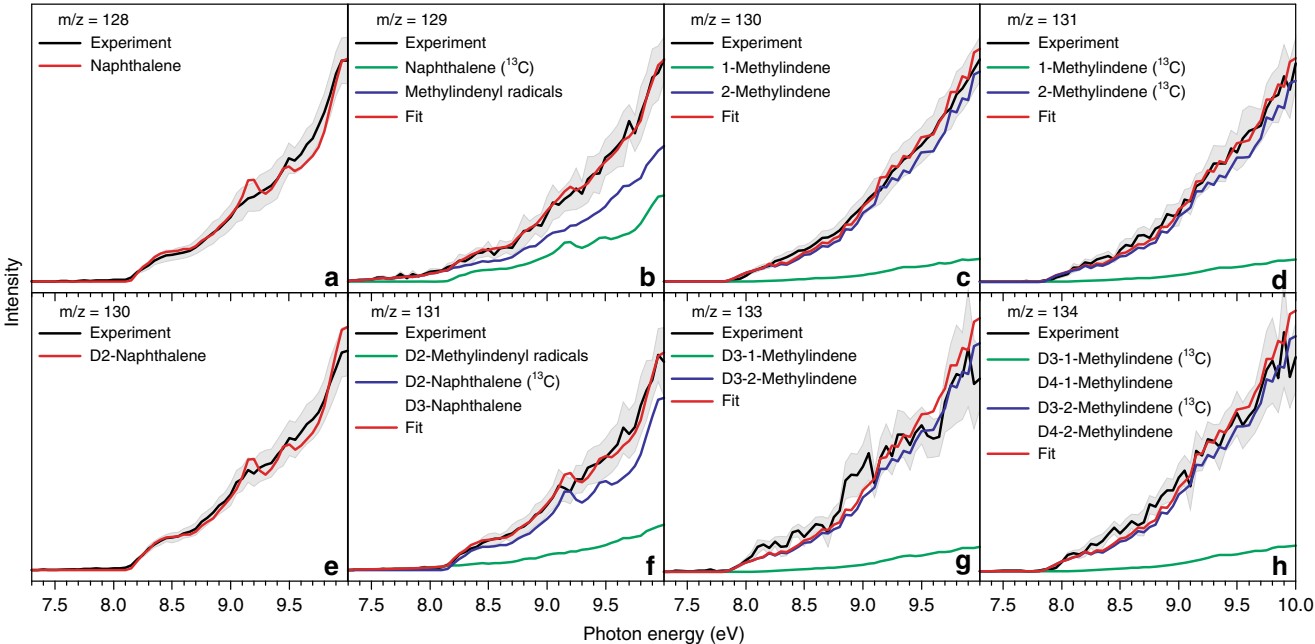

**Fig. 3** PIE curves for different products. **a–d** PIE curves for the reaction of 1-indenyl with methyl and **e–h** PIE curves for the reaction of 1-indenyl with D3-methyl at a reactor temperature of 1425 ± 10 K. Black: experimental PIE curves; blue/green/red: reference PIE curves; in case of multiple contributions to one PIE curve, the red line resembles the overall fit. The error bars consist of two parts: ±10% based on the accuracy of the photodiode and a 1 σ error of the PIE curve averaged over the individual scans. Source data are provided as a Source Data file

overlap after scaling (Fig. 3a, b). A detailed inspection reveals that signal at $m/z = 129$ requires two components to fit the experimental PIE curve: $^{13}$C-naphthalene ($^{13}CC_9H_8$) and the methylindenyl radical ($C_{10}H_9$). The experimentally recorded PIE curve at $m/z = 130$ (Fig. 3c) could be replicated by the linear combination of reference PIE curves of 1- and 2-methylindenes ($C_{10}H_{10}$). The photoionization energy of 2-methylindene is measured to be 7.90 ± 0.05 eV, correlating well with the onset of the experimental signal of $m/z = 130$. Notably, the PIE curves at $m/z = 131$ and 130 are superimposable after scaling, suggesting that signal at $m/z = 131$ (Fig. 3d) solely originates from $^{13}$C 1- and 2-methylindene ($^{13}CC_9H_{10}$). These assignments could be verified in the 1-indenyl —D3-methyl system by fitting the PIE curves at $m/z = 134$, 133, 131, and 130 by $^{13}$C-D3-1-methylindene/$^{13}$C-D3-2-methylindene ($^{13}CC_9H_7D_3$), D3-1-methylindene/D3-2-methylindene ($C_{10}H_7D_3$), D2-methylindenyl/$^{13}$C-D2-naphthalene/D3-naphthalene ($C_{10}H_7D_2$/$^{13}CC_9H_6D_2$/$C_{10}H_5D_3$), and D2-naphthalene ($C_{10}H_6D_2$), respectively (Fig. 3e–h).

## Discussion

Our experiments reveal that the simplest representative of a PAH—naphthalene—can be formed via the reaction of the 1-indenyl radical with the methyl radical following isomerization and loss of two hydrogen atoms at elevated temperatures of 1425 K based on the decomposition temperatures of the radical precursor, which also coincided with the maximum intensities of the products of interest. Electronic structure calculations on the pertinent $C_{10}H_{11}$ and $C_{10}H_{10}$ potential energy surfaces (PESs) (Fig. 1) (Supplementary Figs. 1 and 2) reveal that the reaction is initiated by the barrierless addition of the methyl radical to the C1/C3 and/or C2 carbon atom of 1-indenyl forming 1-methylindene and 2-methyl-2H-indene, respectively. Considering the spin densities at C1/C3 and C2 of 0.52 and 0.20 and the statistical factor of two chemically equivalent carbon atoms C1/C3, the addition to C1/C3 should be favorable compared to C2. These isomers can be interconverted via a barrier of 228 kJ mol$^{-1}$ with respect to 1-methylindene. Besides, 2-methyl-

2H-indene can isomerize to the thermodynamically more stable 2-methylindene isomer by overcoming a barrier of 64 kJ mol$^{-1}$. 1-methylindene and 2-methyl-2H-indene/2-methylindene can undergo decomposition via atomic hydrogen loss in overall endoergic reactions with respect to the separated methyl and 1-indenyl reactants yielding 1- and 2-methylindenyl radicals [i1] and [i2], respectively. Successive hydrogen shifts from the methyl group to the C1 and C2 carbon atoms lead to intermediates [i3] and [i4], respectively, which can each ring-close to an exotic tricyclic intermediate [i5]. Subsequent ring opening forms [i6] bearing the carbon skeleton of naphthalene. The latter decomposes by hydrogen loss to the naphthalene molecule via a tight exit transition state located 26 kJ mol$^{-1}$ above the separated products. All transition states and the reaction endoergicity can be overcome in high temperature environments of combustion flames and in circumstellar envelopes close to the central star, where temperatures can rise to a few thousand K. It is important to highlight that the computational investigation of the 1-indenyl—D3-methyl system supports the aforementioned experimental findings (Fig. 1b). By tracing the deuterium atoms of the D3-methyl reactant, D3-1-methylindene and 2-methyl-2H-indene decompose via atomic hydrogen (H) loss leading to D3-[i1] and D3-[i2] followed by deuterium (D) atom migration to D3-[i3] and D3-[i4], respectively, and ring closure to two distinct intermediates D3-[i5] and D3-[i5]′. Ring opening leads to D3-[i6] and D3-[i6]′, respectively. A deuterium atom (D) loss accompanied by aromatization eventually yields two D2-naphthalene isotopologues; note that hydrogen versus deuterium replacement does not change the shape of the PIE curves within our error limits[17,32–35]; therefore, the naphthalene isotopologues cannot be discriminated experimentally via photoionization. As exhibited in Supplementary Fig. 2, an isomer of naphthalene, benzofulvene, can also be produced. Thus, there is also an indirect route to naphthalene formation, where first benzofulvene is formed, and then attacked by either a hydrogen or deuterium atom, which drives the isomerization to naphthalene. The intensity ratios of the different mass peaks suggest that both the direct channel and this indirect pathway via benzofulvene

contribute to naphthalene formation under our experimental conditions; see the Supplementary Information for more analysis.

Statistical (variable reaction coordinate transition state theory and RRKM-Master Equation) calculations deliver critical temperature- and pressure-dependent rate constants for the 1-indenyl + methyl reaction and successive reactions eventually yielding naphthalene at combustion-relevant pressures and also at nearly zero pressure conditions prevailing in circumstellar envelopes of carbon stars (Supplementary Fig. 3). These calculations fully support our experimental findings of an addition of the methyl radical to the C1/C3 and C2 positions in the entrance channel with addition to C1/C3 being favored by a factor of about 4 in the temperature and pressure range of the reactor. A competition between collisional stabilization of the adducts and the decomposition to 1-/2-methylindenyl plus atomic hydrogen exists ultimately leading to the formation of naphthalene via ring expansion (Supplementary Tables 1 and 2). It is important to highlight that deuterium versus hydrogen isotope scrambling is highly unlikely under our experimental conditions. Isotope scrambling is feasible only via structure [i6], but the hydrogen migration barrier is nearly 42 kJ mol$^{-1}$ higher than the hydrogen elimination barrier and also is entropically less favorable; the calculated rate constant for the hydrogen shift is more than 60 times lower than that for hydrogen atom loss. Thus, scrambling can be practically ruled out.

To conclude, the pathways to naphthalene ($C_{10}H_8$) showcases the prototype of a previously absent class of radical—radical reactions facilitating molecular mass growth processes via ring expansion through the conversion of a five-membered ring to a six-membered ring in PAH radicals. This mechanism provides a pathway to a novel high temperature formation of larger PAHs leading eventually to soot and carbonaceous grains in deep space. The elementary reactions between the methyl radical and (annulated) cyclopentadienyl radicals deliver a versatile ring expansion mechanism beyond 1-indenyl radicals converting previously postulated non-reactive end members of aromatic growth processes, which carry five-membered rings, to planar aromatic systems, such as anthracene ($C_{14}H_{10}$), phenanthrene ($C_{14}H_{10}$), eventually culminating in graphene-type PAHs (Supplementary Figs. 10 and 11); these two-dimensional nanostructures resemble graphene-type layers and define molecular building blocks of graphite as detected in the L3 chondrite Khohar[36] and in Murchison[37]. The mechanism outlined here will be crucial in interpreting chemical models of combustion processes[5] and of carbon-rich circumstellar environments[4], help explain observation of graphitized carbon grains as detected in carbonaceous chondrites[36–40], and enrich our knowledge of the formation and evolution of carbonaceous matter in the galaxy.

## Methods

**Experimental.** The experiments were carried out at the Chemical Dynamics Beamline (9.0.2) of the Advanced Light Source utilizing a resistively heated silicon–carbide (SiC) chemical reactor interfaced to a molecular beam apparatus operated with a Wiley–McLaren reflectron time-of-flight mass spectrometer (Re-TOF-MS)[17,31,32,34,41,42]. The chemical reactor mimics the high temperature conditions present in combustion flames and in circumstellar envelopes of carbon stars. The 1-indenyl radical ($C_9H_7^{\bullet}$) and the methyl/D3-methyl radical ($CH_3^{\bullet}/CD_3^{\bullet}$) were prepared in situ via pyrolysis of the 1-bromoindene precursor ($C_9H_7Br$; synthesized in this work, Supplementary Note 1) and acetone/D6-acetone ($CH_3COCH_3/CD_3COCD_3$, Sigma-Aldrich, 99.9%). The reactants were seeded in helium carrier gas (0.394 ± 0.005 atm) and introduced into a resistively heated silicon–carbide tube (SiC) with the temperature of 1425 ± 10 K monitored using a Type-C thermocouple. The products formed in the reactor were expanded supersonically and passed through a 2 mm diameter skimmer located 10 mm downstream of the pyrolytic reactor and enter into the main chamber, which houses the Re-TOF-MS. The quasi-continuous tunable vacuum ultraviolet (VUV) light from the Advanced Light Source intercepted the neutral supersonic molecular beam perpendicularly in the extraction region of RE-TOF-MS. VUV

single photon ionization is essentially a fragment-free ionization technique and hence is characterized as a soft ionization method[43]. The ions formed via photoionization are extracted and detected by a multichannel plate detector. Photoionization efficiency (PIE) curves, which report ion counts as a function of photon energy ranged from 7.30 to 10.00 eV, with a step interval of 0.05 eV at a well-defined mass-to-charge ratio ($m/z$), were produced by integrating the signal recorded at the specific $m/z$ for the species of interest. Reference (blank) experiments were also conducted by expanding helium carrier gas into the resistively heated SiC tube with only seeded 1-bromoindene. To identify the products of interest observed in this work, PIE calibration curves for 1-methylindene (BOCSCI) and 2-methylindene (Sigma-Aldrich) were also collected within the same experimental setup.

**Computational.** Stationary points on the $C_{10}H_{10}$ potential energy surface (PES) of interest were obtained using density functional theory (DFT) optimizations at the B3LYP/6-311G(d,p)[44,45] level. The DFT B3LYP/6-311G(d,p) method was also used to compute vibrational frequencies and zero-point energy (ZPE) corrections. The energies of the stationary points were refined with the explicitly correlated coupled clusters CCSD(T)-F12/cc-pVTZ-f12[46,47] method. The DFT calculations were carried out using the Gaussian 09 program package[48]. The coupled clusters calculations were performed using the MOLPRO 2010 program[49].

For the reactions with distinct barriers, energy- and angular momentum-resolved (E,J-resolved) rate constants of the unimolecular reactions were calculated using Rice–Ramsperger–Kassel–Marcus (RRKM) theory[50]. Namely, the number of states for a transition state and the density of states for the related local minima were generally calculated using the rigid-rotor, harmonic-oscillator (RRHO) model. Low-frequency normal modes corresponding to internal rotation were considered as hindered rotors, which replaced the corresponding harmonic oscillators in RRHO. Hindered rotor potentials were scanned using the B3LYP/6-311G(d,p) level of theory. The barrierless unimolecular decompositions and reverse bimolecular associations were treated using the multifaceted implementation of Variable Reaction Coordinate Transition State Theory (VRC-TST)[51]. For distances less than 6 Å (short-range), the pivot points were placed at 0.25 and 0.5 Å apart from the interacting carbons of methyl and aromatic moieties, respectively, and perpendicular to their molecular planes. The multireference character of a wave function for radical–radical interactions was accounted for using the second-order perturbation theory CASPT2 method[52,53] with the cc-pvdz basis set. To avoid discontinuities in the potential describing the interaction between the radicals, one of which is resonantly stabilized, it is required to include the delocalized radical orbitals in the active space. CASPT2 calculations were carried out using the (10e,10o) active space which includes not only the delocalized radicals but the complete π system plus the orbital of an incipient bond. The minimum energy path (MEP) geometry relaxation corrections were obtained with complete active space CASSCF (10e,10o) method[54]. The complete basis set (CBS) correction[55] was applied based on the CASPT2(10e,10o)/cc-pVnZ ($n$ = D, T, Q) energies of the unrelaxed MEP structures using the following scheme (Eqs. (1)–(3)):

$$\Delta E[\text{pVTZ}] = E_{\text{rigid}}[\text{CASPT2}(10e, 10o)/cc-VTZ] \\ - E_{\text{rigid}}[\text{CASPT2}(10e, 10o)/cc-VDZ] \quad (1)$$

$$\Delta E[\text{pVQZ}] = E_{\text{rigid}}[\text{CASPT2}(10e, 10o)/cc-VQZ] \\ - E_{\text{rigid}}[\text{CASPT2}(10e, 10o)/cc-VTZ] \quad (2)$$

$$\Delta E[\text{CBS}] = \Delta E[\text{pVQZ}] + 0.69377 \\ \times (\Delta E[\text{pVQZ}] - \Delta E[\text{pVTZ}]) \quad (3)$$

where $E_{\text{rigid}}$ are single point energies of interacting fragments when brought into a particular dividing surface configuration without geometry relaxation relative to the energy of these fragments been at infinite separation.

The temperature and pressure-dependent phenomenological rate constants were calculated by solving the one-dimensional master equation (ME)[56] using the MESS program package[57]. The collisional energy transfer and Lennard-Jones parameters for ME were taken from the previous study of the $C_9H_x$/Ar systems[23]. Namely, the Lennard-Jones parameters were (ε/cm$^{-1}$, σ/Å) = (390, 4.46) and $n$ = 0.62, $\alpha_{300}$ = 424 cm$^{-1}$ were used in the "exponential down" model[58] of the collisional energy transfer for the temperature dependence of the range parameter $\alpha$ for the deactivating wing of the energy transfer function $\alpha(T) = \alpha_{300}(T/300 \text{ K})^n$. The parameters in the form of MESS input file are provided in Supplementary Note 2.

## Data availability

The data that support the plots within this paper and other findings of this study are available from the corresponding author upon reasonable request. Source data underlying Figs. 2, 3 and Supplementary Figs. 3, 4, 5, 6, and 7 are provided as a Source Data file.

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

## Acknowledgements

This work was supported by the US Department of Energy, Basic Energy Sciences DE-FG02-03ER15411 (experimental studies) and DE-FG02-04ER15570 (computational

studies; synthesis of 1-bromoindene) to the University of Hawaii (UH) and Florida International University (FIU), respectively. W.L., B.X. and M.A. are supported by the Director, Office of Science, Office of Basic Energy Sciences, of the U.S. Department of Energy under Contract No. DE-AC02-05CH11231, through the Gas Phase Chemical Physics program of the Chemical Sciences Division. The ALS is supported under the same contract. A.H.H. acknowledges support from FIU for a Presidential Fellowship.

## Author contributions

R.I.K. designed the experiment; L.Z., B.X and W.L. carried out the experimental measurements; M.A. supervised the experiment; L.Z. and R.I.K. performed the data analysis; A.N.M. and A.M.M. carried out the theoretical analysis; A.H.H. and S.F.W. synthesized the compounds, A.M.M., and M.A. discussed the data; R.I.K., A.M.M. and L.Z. wrote the paper.

## Additional information

**Competing interests:** The authors declare no competing interests.

