## [Peer Review File · Nature Communications]

Reviewers' comments:

Reviewer #1 (Remarks to the Author):

This manuscript reports experimental measurements on the methyl radical + 1-indenyl radical reaction around 1400 K, showing it makes comparable amounts of naphthalene (unexpected) along with the expected methylindene. The authors include a very interesting isotope labeling experiment which indicates that two of the deuteriums from CD₃ end up on the naphthalene, indicating D atom transfer from the methyl group to a carbon in the ring. The authors also verify that the C₁₀H₈ product is naphthalene (e.g. not benzofulvene) from the PIE spectrum. The authors suggest a unimolecular reaction mechanism consistent with their observations, and back it up with some high level quantum chemistry calculations that show it is plausible. The mechanism is analogous to one that the authors studied previously in Ref. 24 (cyclopentadienyl + CH₃). The work is interesting and definitely worth publishing.

However I don't think the authors have completely proven the mechanism, at least not to my satisfaction. Prior work in Refs 19 and 22 suggest an alternative bimolecular mechanism, where the dominant product methylindenyl first loses an H atom from the methyl group forming benzofulvene, either as a unimolecular beta scission reaction or by H-abstraction being attacked by another radical. Then an H atom adds to benzofulvene inducing isomerization to benzocyclohexadienyl radical, and that radical loses an H atom to form naphthalene. The deuterium labeling experiments appear to favor the unimolecular mechanism, but I think that is not definitive, since I think the H/D can move around the benzocyclohexadienyl radical with low barriers, and loss of H is favored over D because of zero point energy effects.

It would make this manuscript stronger if the authors could compute the rate of that alternative bimolecular route to show that it cannot be correct. But of course that is a bit tricky since then the authors would need to estimate the concentration of radicals and H atoms inside their reactor, and perhaps even consider the possibility that wall reactions matter. Another way to make the manuscript stronger would be to quantitatively predict the rates of the pathway shown in Fig. 2 and show that it is consistent with the observed product ratio. Yet another course of action would be to add a sentence saying that the present work, particularly the observation that 2 D's are retained in the naphthalene, suggests the unimolecular pathway in Fig. 1 is dominant, but that one cannot completely exclude bimolecular reactions as suggested in Refs. 19 and 22, and also computed in Ref. 24.

Reviewer #2 (Remarks to the Author):

This manuscript describes a combined experimental and computational investigation of a new pathway identified by the authors to naphthalene, involving methyl + indenyl radical recombination, ring expansion, and loss of two H-atoms to produce naphthalene. The combination of a pyrolysis microreactor with tunable VUV photoionization enables the authors to follow the steps via mass spectrometry + photoionization efficiency scans that identify the carrier of the C₁₀H₈ or C₁₀H₆D₂ as due to naphthalene. The evidence for the pathway is firmly established by the deuterium substitution, the PIE onset, and backed up by theory. The authors argue that this pathway is quite general as a tool for growing larger PAH, and make bold claims along those lines. Time will tell whether this is a facile pathway or not, since no assessment is made in this work about the efficiency of this process relative to other competitors in flames and carbon-rich stars. Nevertheless, the chemistry is interesting to a broad audience of readers of the journal, poses a new pathway that needs to be considered further, and makes a strong experimental and computational case for the pathway. The paper is well-written. At times the written description of the chemistry is difficult to follow, but this is simply an aspect of the work that is unavoidable -- following the peaks in the mass spectra, the subtle differences in PIE curves, and the multi-step reaction pathway. I only have a couple of minor comments:

1. Why is mass 134 almost as big as 133 in the CD₃ mass spectra of Figure 1? Shouldn't it be 9%?

2. Are reference spectra for all the possible methyl indenenes available (not just 1- and 2-methyl)?
3. How was the temperature of 1400 K chosen for the study? Is it the temperature at which both radicals are formed cleanly?

No assessment of pathway relative to others.

Reviewer #3 (Remarks to the Author):

The authors present a thorough investigation of the radical-radical reaction between indenyl and methyl, remarkably leading to naphthalene. The photoionization experiments identify indenyl and methyl as pyrolysis products of bromide and acetone precursors, respectively, as well as the methyl-indene intermediate and the naphthalene final product. The most relevant contribution of the work is in fact the experimental demonstration of the plausibility of the ring expansion route for this system, converting the five-membered ring of indene to the six-membered ring of naphthalene. The whole reaction process constitutes a valuable benchmark to understand PAH growth in different environments, complementing other possible pathways (e.g. HACA routes). To this respect, it would be interesting if the authors could make any assessment about the potential formation of indenyl dimers in their experiments (do they observe any signal at $m/z=230$?), as this could constitute the initial step of further routes of PAH growth.

Perhaps, a weak aspect of the paper is that the reaction under study as such is not a novelty from a conceptual point of view. It has been described in detail previously by the authors (Mebel et al. 2016, ref.20 of the paper). For instance, the potential energy diagrams reported in this paper to explain the transformation of methyl-indenyl to naphthalene (Fig. S2) was published in essentially the same form in ref. 20. Based on that prediction, the results of the study could to some extent be anticipated and part of the analysis outlined in the present paper is somewhat redundant.

Despite such lack of novelty, the experimental proof of the reaction and the extended computational analysis provided in this investigation are certainly meritorious and deserve publication in a high impact journal.

Reviewer #4 (Remarks to the Author):

The present study is a fine experimental and theoretical work, but using the same methodology as the paper published in 2018 in the Journal of Physical Chemistry letters by a team with many common members with the present one (ref 26 of the present paper). This paper was also on a reaction leading to naphthalene. By the way, I am not sure that the word "synthesis", which is used several times in the paper, is the appropriate one when studying reaction kinetics. In addition, it is not clear how the proposed reaction, the combination of methyl and indenyl radicals, can be so important. The way of formation of the bicyclic indenyl radicals does not seem to be so easy either in combustion or astrochemical environments. The radical-radical reaction, which is to be studied for naphthalene formation, should rather be the recombination of cyclopentadienyl radicals. While these last radicals can easily be obtained from benzene, this recombination is not even mentioned in the paper. Note that a paper on this subject has recently been published: "Pressure dependent kinetic analysis of pathways to naphthalene from cyclopentadienyl recombination" by Long et al., *Combustion and Flame*, 187 (2018) 247-256. Consequently, I am not sure that the novelty and importance of this paper are high enough for it to be published in "Nature Communications".

Reviewer #1

This manuscript reports experimental measurements on the methyl radical + 1-indenyl radical reaction around 1400 K, showing it makes comparable amounts of naphthalene (unexpected) along with the expected methylindene. The authors include a very interesting isotope labeling experiment which indicates that two of the deuteriums from CD₃ end up on the naphthalene, indicating D atom transfer from the methyl group to a carbon in the ring. The authors also verify that the C₁₀H₈ product is naphthalene (e.g. not benzofulvene) from the PIE spectrum. The authors suggest a unimolecular reaction mechanism consistent with their observations, and back it up with some high level quantum chemistry calculations that show it is plausible. The mechanism is analogous to one that the authors studied previously in Ref. 24 (cyclopentadienyl + CH₃). The work is interesting and definitely worth publishing.

Thank you.

However I don't think the authors have completely proven the mechanism, at least not to my satisfaction. Prior work in Refs 19 and 22 suggest an alternative bimolecular mechanism, where the dominant product methylindenyl first loses an H atom from the methyl group forming benzofulvene, either as a unimolecular beta scission reaction or by H-abstraction being attacked by another radical. Then an H atom adds to benzofulvene inducing isomerization to benzocyclohexadienyl radical, and that radical loses an H atom to form naphthalene. The deuterium labeling experiments appear to favor the unimolecular mechanism, but I think that is not definitive, since I think the H/D can move around the benzocyclohexadienyl radical with low barriers, and loss of H is favored over D because of zero point energy effects. It would make this manuscript stronger if the authors could compute the rate of that alternative bimolecular route to show that it cannot be correct. But of course that is a bit tricky since then the authors would need to estimate the concentration of radicals and H atoms inside their reactor, and perhaps even consider the possibility that wall reactions matter. Another way to make the manuscript stronger would be to quantitatively predict the rates of the pathway shown in Fig. 2 and show that it is consistent with the observed product ratio. Yet another course of action would be to add a sentence saying that the present work, particularly the observation that 2 D's are retained in the naphthalene, suggests the unimolecular pathway in Fig. 1 is dominant, but that one cannot completely exclude bimolecular reactions as suggested in Refs. 19 and 22, and also computed in Ref. 24.

Thank you for this comment and providing us three options how to address this. We feel that the best answer is to proceed with option 3. Also, please note that D/H isotope scrambling suggested is highly unlikely. Isotope scrambling is possible only via structure i6, but the H migration barrier is almost 42 kJmol⁻¹ higher than the H elimination barrier and also is entropically less favorable; the calculated rate constant for H shift at the reactor's temperature is more than 60 times lower than that for H loss. Thus, the scrambling can be practically ruled out. Therefore, we have added the following sections. Due to space limitations, one section we placed in the main manuscript, the second section in the Supplementary Material section:

Main Manuscript:

It is important to highlight that deuterium versus hydrogen isotope scrambling is highly unlikely under our experimental conditions. Isotope scrambling is feasible only via structure i6, but the hydrogen migration barrier is nearly 42 kJmol^{-1} higher than the hydrogen elimination barrier and also is entropically less favorable; the calculated rate constant for the hydrogen shift is more than 60 times lower than that for hydrogen atom loss. Thus, scrambling can be practically ruled out.

Supplementary Material:

The calculated branching ratios of benzofulvene versus naphthalene at the conditions of our reactor are nearly equal to 4 to 1. Therefore, we expected the bimolecular mechanism, i.e. deuterium (D) loss from methylindenyl to eventually form benzofulvene followed by H-assisted isomerization of the latter to naphthalene, to be the most favorable mechanism. This mechanism is consistent with the isotope-labeling result if benzofulvene re-reacts with D and isomerizes to naphthalene through such D-assisted isomerization. However, it is not consistent if benzofulvene reacts with H rather than with D - then the reaction would produce naphthalene with only one D incorporated ($m/z = 129$). The concentrations of hydrogen and deuterium are expected to be roughly equal because the dissociation of 1- or 2-methylindene produces hydrogen and the dissociation of D3-methylindenyl radicals produces deuterium. Therefore, technically these calculations show that 1/5 of naphthalene is produced via the unimolecular mechanism containing two deuterium atoms; 2/5 of naphthalene is likely produced via the bimolecular mechanism via a deuterium-assisted isomerization of benzofulvene containing also two deuterium atoms; finally, 2/5 of naphthalene are formed via a hydrogen-assisted isomerization of benzofulvene and contain only one deuterium. Therefore, qualitatively speaking, the intensity of the ion counts at $m/z = 130$ are expected to be higher than at $m/z = 129$, which agrees with our experimental findings.

Reviewer #2

This manuscript describes a combined experimental and computational investigation of a new pathway identified by the authors to naphthalene, involving methyl + indenyl radical recombination, ring expansion, and loss of two H-atoms to produce naphthalene. The combination of a pyrolysis microreactor with tunable VUV photoionization enables the authors to follow the steps via mass spectrometry + photoionization efficiency scans that identify the carrier of the C₁₀H₈ or C₁₀H₆D₂ as due to naphthalene. The evidence for the pathway is firmly established by the deuterium substitution, the PIE onset, and backed up by theory.

Thank you.

The authors argue that this pathway is quite general as a tool for growing larger PAH, and make bold claims along those lines. Time will tell whether this is a facile pathway or not, since no assessment is made in this work about the efficiency of this process relative to other competitors in flames and carbon-rich stars.

Thank you. We agree.

Nevertheless, the chemistry is interesting to a broad audience of readers of the journal, poses a new pathway that needs to be considered further, and makes a strong experimental and computational case for the pathway. The paper is well-written. At times the written description of the chemistry is difficult to follow, but this is simply an aspect of the work that is unavoidable -- following the peaks in the mass spectra, the subtle differences in PIE curves, and the multi-step reaction pathway.

Thank you.

I only have a couple of minor comments:

1. Why is mass 134 almost as big as 133 in the CD₃ mass spectra of Figure 1? Shouldn't it be 9%?

Please note that we cannot differentiate between ¹³C-D₃-methylindene and D₄-methylindene. Based on the isotope ratio and the experimental findings, this would suggest that we are making copious amounts of D₄-methylindene, which means that a deuterium atom is also substituting a hydrogen atom in the ring as described in Figure S9 in the Supplementary Material. Here, the reaction of indenyl plus deuterium forming D₁-indene followed by decomposition to D₁-indenyl plus atomic hydrogen is exoergic because the C-H bond is weaker than the C-D bond due to the effect of zero-point vibrational energy. This process is followed by the reaction of D₁-indenyl plus D₃-methyl to give D₄-methylindene ($m/z = 134$).

We added this section in the Supplementary Material.

2. Are reference spectra for all the possible methyl indenenes available (not just 1- and 2-methyl)?

Reference spectra for all the possible isomers of methyl indenenes are not available but we did not have to measure them because, according to the theoretical calculation, 1- and 2-methylindenenes are the key 'methylated' products generated in the methyl – 1-indenyl reaction. A linear combination of the PIE curves of both isomers fit the experimental PIE curve very well.

3. How was the temperature of 1400 K chosen for the study? Is it the temperature at which both radicals are formed cleanly?

We chose this temperature based on the best decomposition temperatures of the radical precursor, which also coincided with the best intensities of the products of interest. As stated in the manuscript, to make sure that no other radicals affect the reaction, reference (blank) experiments were also conducted by expanding helium carrier gas into the resistively-heated SiC tube with only seeded 1-bromoindene, acetone or D6-acetone, respectively. PIE fits and mass spectra show that signal at $m/z = 115$ in 1-bromoindene/helium is attributed only to 1-indenyl; signal at $m/z = 15$ in acetone/helium is attributed to methyl and signal at $m/z = 18$ in D6-acetone/helium is attributed to D3-methyl revealing that the radicals are cleanly formed.

Reviewer #3

The authors present a thorough investigation of the radical-radical reaction between indenyl and methyl, remarkably leading to naphthalene. The photoionization experiments identify indenyl and methyl as pyrolysis products of bromide and acetone precursors, respectively, as well as the methyl-indene intermediate and the naphthalene final product. The most relevant contribution of the work is in fact the experimental demonstration of the plausibility of the ring expansion route for this system, converting the five-membered ring of indene to the six-membered ring of naphthalene. The whole reaction process constitutes a valuable benchmark to understand PAH growth in different environments, complementing other possible pathways (e.g. HACA routes).

Thank you.

To this respect, it would be interesting if the authors could make any assessment about the potential formation of indenyl dimers in their experiments (do they observe any signal at $m/z=230?$), as this could constitute the initial step of further routes of PAH growth.

We checked the mass spec for the indenyl dimer as requested; we do not see any evidence.

Perhaps, a weak aspect of the paper is that the reaction under study as such is not a novelty from a conceptual point of view. It has been described in detail previously by the authors (Mebel et al. 2016, ref.20 of the paper). For instance, the potential energy diagrams reported in this paper to explain the transformation of methyl-indenyl to naphthalene (Fig. S2) was published in essentially the same form in ref. 20. Based on that prediction, the results of the study could to some extent be anticipated and part of the analysis outlined in the present paper is somewhat redundant. Despite such lack of novelty, the experimental proof of the reaction and the extended computational analysis provided in this investigation are certainly meritorious and deserve publication in a high impact journal.

It is important to highlight that – as stated in the manuscript – we conducted for the very first time experimentally a high temperature reaction between two astrochemically and combustion chemically relevant radicals leading eventually to the formation of naphthalene ($C_{10}H_8$) - *the* prototype PAH carrying two fused benzene rings. Therefore, we provided compelling **evidence** that this reaction does lead to naphthalene. In science, many predictions can be made from theory, in particular, by electronic structure calculations, but also very often these predictions were shown to be incorrect in the past. For instance, the HACA mechanism was considered as a dominant pathway for PAH formation for over three decades, however, recently it has been shown experimentally that other pathways such as HAVA and radical-radical reactions (as demonstrated here) can also play crucial roles. Therefore, experimental evidence is key to critically advance our knowledge of reaction systems relevant to astrochemistry and combustion sciences.

Reviewer #4

The present study is a fine experimental and theoretical work, but using the same methodology as the paper published in 2018 in the Journal of Physical Chemistry letters by a team with many common members with the present one (ref 26 of the present paper). This paper was also on a reaction leading to naphthalene.

Many reactions have been proposed to lead to naphthalene, but a *radical-radical* reaction pathway has never been before shown experimentally to form naphthalene due to previously insurmountable experimental difficulties. In fact, since naphthalene is *the* prototype PAH carrying two fused benzene rings, it is the goal of this team is to unravel, experimentally and theoretically, *all* possible pathways to naphthalene, so that kinetic models could evaluate relative contributions of different pathways to the PAH growth under various conditions in combustion and in astrochemistry.

The way of formation of the bicyclic indenyl radicals does not seem to be so easy either in combustion or astrochemical environments.

We cannot agree with the reviewer on this point. No reference is made to the literature or evidence provided to back up this statement. On the contrary, reference 22 reviews and compares various pathways for the formation of indene in great detail.

The radical-radical reaction, which is to be studied for naphthalene formation, should rather be the recombination of cyclopentadienyl radicals. While these last radicals can easily obtained from benzene, this recombination is not even mentioned in the paper. Note that a paper on this subject has recently been published: "Pressure dependent kinetic analysis of pathways to naphthalene from cyclopentadienyl recombination" by Long et al., Combustion and Flame, 187 (2018) 247-256.

While we agree with the reviewer that cyclopentadienyl recombination can also contribute to the formation of naphthalene and added this possible pathway in the introduction, the subject of this particular paper is a *different reaction*. The $C_5H_5 + C_5H_5$ reaction has never been studied experimentally and can certainly be conducted with our approach in the future. . The pathways, which are verified experimentally, can be confidentially included in physically justified kinetic models and those will resolve relative contributions of different pathways to the formation of naphthalene in various conditions.

In addition, it is not clear how the proposed reaction, the combination of methyl and indenyl radicals, can be so important.

The importance was clearly laid out in our manuscript (introduction, conclusion).Referee 1-3 gave a very positive assessment of our case for this.

By the way, I am not sure that the word “synthesis”, which is used several times in the paper, is the appropriate one when studying reaction kinetics.

We changed synthesis to formation in the manuscript.

REVIEWERS' COMMENTS:

Reviewer #1 (Remarks to the Author):

This is basically a good manuscript reporting a very interesting experiment on a potentially important reaction. However, there are three aspects of the manuscript that need alteration:

1) The manuscript reads as if the experiments are 100% consistent with the direct reaction $\text{CH}_3 + \text{indenyl}$ to naphthalene being the dominant source of naphthalene, but a closer look at the data indicates that the multistep route involving a benzofulvene intermediate reacting with H or D atoms is comparably important. This is explained in the Supporting Information, but some of these needs to be stated in the main article text. I suggest adding these two sentence: "As discussed in the literature, there is also an indirect route to naphthalene formation, where first benzofulvene is stabilized, and then attacked by either H or D atom, which drives its isomerization to naphthalene. The intensity ratios of the different mass peaks we measure suggest that both the direct channel and this indirect pathway via benzofulvene + H or D both contribute significantly to naphthalene formation at our conditions; see the Supporting Information for more analysis."

2) The apparent fact that the reacting systems includes H and D atoms at high enough concentrations that they make the benzofulvene pathway run suggest that there could be significant H/D scrambling by reactions like $\text{H} + \text{C}_{10}\text{H}_6\text{D}_2 = \text{D} + \text{C}_{10}\text{H}_7\text{D}_1$. These need to be discussed in the Supporting Information. If they are significant they also need to be mentioned in the main text since they might confuse the interpretation of the isotope experiments. Do you know if the reaction I wrote would run in the forward direction or the reverse direction at your reaction conditions?

3) The abstract suggests that this is the very first radical + radical reaction forming naphthalene to be observed experimentally. But in fact, the $\text{C}_5\text{H}_5 + \text{C}_5\text{H}_5$ to naphthalene reaction was previously observed multiple times, most importantly in the very direct and clean experiment of Knyazev and Popov, JPCA 2015. Please cite Knyazev & Popov in the introduction. An informed reader reading the current version of the Abstract would assume you are discussing $\text{C}_5\text{H}_5 + \text{C}_5\text{H}_5$, and would be wondering "what are they going to show about this reaction that Knyazev didn't already show?". This is not what you want. Please change the abstract to mention that this manuscript reports observation of the reaction $\text{CH}_3 + \text{indenyl}$ forming naphthalene, and that it includes isotope labeling experiments.

Reviewer #3 (Remarks to the Author):

I have re-evaluated the paper in the light of the comments of the four reviewers, and the corresponding responses of the authors. The authors account for the main issues raised by the reviewers. There is agreement among the reviewers on the high quality of the reported research and the relevance of the results. It certainly deserves publication in a high impact journal.

In my view, the key question is whether the paper provides enough scientific novelty, given the stringent requirements for publication in Nature Communications. As pointed out in the review report, the reaction under study was anticipated in a previous paper, where the reaction pathway was described in detail. The authors stress in their response the importance of the experimental evidence provided in the paper for the predicted reaction. Whereas I can only agree with their statements, I am not convinced that the impact of the paper in the field will reach the outstanding level demanded for publication in Nature Communications.

Reviewer #4 (Remarks to the Author):

The paper has been well reviewed and can be accepted.

Reviewer #1:

This is basically a good manuscript reporting a very interesting experiment on a potentially important reaction. However, there are three aspects of the manuscript that need alteration:

1) The manuscript reads as if the experiments are 100% consistent with the direct reaction $\text{CH}_3 + \text{indenyl}$ to naphthalene being the dominant source of naphthalene, but a closer look at the data indicates that the multistep route involving a benzofulvene intermediate reacting with H or D atoms is comparably important. This is explained in the Supporting Information, but some of these needs to be stated in the main article text. I suggest adding these two sentence: “As discussed in the literature, there is also an indirect route to naphthalene formation, where first benzofulvene is stabilized, and then attacked by either H or D atom, which drives its isomerization to naphthalene. The intensity ratios of the different mass peaks we measure suggest that both the direct channel and this indirect pathway via benzofulvene + H or D both contribute significantly to naphthalene formation at our conditions; see the Supporting Information for more analysis.”

Thanks for the suggestion. We added the following sentences into the manuscript:

As exhibited in Supplementary Figure 2, an isomer of naphthalene, benzofulvene, can also be produced. Thus, there is also an indirect route to naphthalene formation, where first benzofulvene is formed, and then attacked by either a hydrogen or deuterium atom, which drives the isomerization to naphthalene. The intensity ratios of the different mass peaks suggest that both the direct channel and this indirect pathway via benzofulvene contribute to naphthalene formation under our experimental conditions; see the Supplementary Information for more analysis.

2) The apparent fact that the reacting systems includes H and D atoms at high enough concentrations that they make the benzofulvene pathway run suggest that there could be significant H/D scrambling by reactions like $\text{H} + \text{C}_{10}\text{H}_6\text{D}_2 = \text{D} + \text{C}_{10}\text{H}_7\text{D}_1$. These need to be discussed in the Supporting Information. If they are significant they also need to be mentioned in the main text since they might confuse the interpretation of the isotope experiments. Do you know if the reaction I wrote would run in the forward direction or the reverse direction at your reaction conditions?

The experimental measurements suggest that the benzofulvene intermediate is isomerized to naphthalene under our conditions. In this process, there might be H/D scrambling, but it will not affect the experiment results.

We already had the description in the Supplementary Information: “Therefore, technically these calculations show that 1/5 of naphthalene is produced via the unimolecular mechanism containing two deuterium atoms; 2/5 of naphthalene is likely produced via the bimolecular mechanism via a deuterium-assisted isomerization of benzofulvene containing also two deuterium atoms; finally, 2/5 of naphthalene are formed via a hydrogen-assisted isomerization of benzofulvene and contain only one deuterium.” We further added there: “Further H/D isotope scrambling involving naphthalene itself, like $\text{H} + \text{C}_{10}\text{H}_6\text{D}_2$ (naphthalene) \rightleftharpoons $\text{D} + \text{C}_{10}\text{H}_7\text{D}_1$ (naphthalene) is favorable in reverse direction because a C-D bond is stronger than a C-H bond due to the effect of ZPE and thus, the H/D isotope scrambling will additionally boost the $m/z = 130$ peak over $m/z = 129$.”

3) The abstract suggests that this is the very first radical + radical reaction forming naphthalene to be observed experimentally. But in fact, the $\text{C}_5\text{H}_5 + \text{C}_5\text{H}_5$ to naphthalene reaction was previously observed multiple times, most importantly in the very direct and clean experiment of Knyazev and Popov, JPCA 2015. Please cite Knyazev & Popov in the introduction. An informed reader reading

the current version of the Abstract would assume you are discussing $C_5H_5 + C_5H_5$, and would be wondering "what are they going to show about this reaction that Knyazev didn't already show?". This is not what you want. Please change the abstract to mention that this manuscript reports observation of the reaction $CH_3 + indenyl$ forming naphthalene, and that it includes isotope labeling experiments.

We revised this in the Abstract and cited the reference in the Introduction. This sentence reads now:

“We present a mechanism through laboratory experiments and computations revealing how the prototype PAH - naphthalene - can be efficiently formed via a rapid 1-indenyl radical – methyl radical reaction.”

Report reviewer #2:

I have reviewed the revised version of this manuscript. The issues I raised in my previous review have been addressed in part, but see below. There is no need for further review.

(1) In response to my query about mass 134 being almost as large as 133 in the CD3 mass spectra of Figure 1, the authors postulate that 'copious amounts of D4-methylindene must be produced by D-atom substitution in the ring. The authors point us to Figure S9 in supplementary material. The thing that still concerns me is that the isotope dependence is used as evidence for formation of naphthalene in a single radical-radical recombination, but in the processes in Figure S9, a series of bimolecular reactions between radicals and precursor leads to the D4-methylindene.

The reaction of 1-indenyl plus deuterium forming D1-indene followed by decomposition to D1-indenyl plus atomic hydrogen is exoergic because the C-H bond is weaker than the C-D bond due to the effect of zero-point vibrational energy. The relatively strong signals of the deuterated indenyl radicals (Supplementary Figures 4c and 7) also suggest the existence of D1-1-indenyl radical, one of the precursors of D4-methylindene. These bimolecular reactions show the potential formation processes for deuterated indenyl radicals, and they do not go against the fact that naphthalene is formed via a 1-indenyl - methyl recombination followed by H-loss steps. Just like the formation of D3-methylindene via the reaction of 1-indenyl with D3-methyl, D4-methylindene is produced via the reaction of D1-1-indenyl with D3-methyl. In conclusion, the major source of paramount 133 and 134 peaks may not be mainly from bimolecular reactions, but they should originate from different indenyl precursors.

We added the previous paragraph in the 2nd paragraph of Page S20 in Supplementary Information.

(2) I'm not sure that an argument that assumes that the theory is correct and therefore there's no need to consider other possibilities is an especially strong argument.

It is not only the theoretical calculations, but the excellent fit of the experimental PIE with the linear combination of 1- and 2-methylindene reference PIE curves that suggest **theoretically and experimentally** that there should be only two isomers for methylindene.

(3) The authors need to add a statement to the manuscript to explain the 1400 K temperature.

We added this sentence in the manuscript:

“Our experiments reveal that the simplest representative of a PAH – naphthalene - can be formed via the reaction of the 1-indenyl radical with the methyl radical following isomerization and loss of two hydrogen atoms at elevated temperatures of 1,400 K based on the decomposition temperatures of the radical precursor, which also coincided with the maximum intensities of the products of interest.”

Reviewer #3:

I have re-evaluated the paper in the light of the comments of the four reviewers, and the corresponding responses of the authors. The authors account for the main issues raised by the reviewers. There is agreement among the reviewers on the high quality of the reported research and the relevance of the results. It certainly deserves publication in a high impact journal.

In my view, the key question is whether the paper provides enough scientific novelty, given the stringent requirements for publication in Nature Communications. As pointed out in the review report, the reaction under study was anticipated in a previous paper, where the reaction pathway was described in detail. The authors stress in their response the importance of the experimental evidence provided in the paper for the predicted reaction. Whereas I can only agree with their statements, I am not convinced that the impact of the paper in the field will reach the outstanding level demanded for publication in Nature Communications.

We respectfully disagree.

Reviewer #4:

The paper has been well reviewed and can be accepted.

Thank you.